The homeodomain factor Gbx1 is required for locomotion and cell specification in the dorsal spinal cord

Meziane Hamid 1
Fraulob Valérie 2
Riet Fabrice 1
Krezel Wojciech 2
Selloum Mohammed 1
Geffarth Michaela 3
Acampora Dario 4 5
Hérault Yann 1
Simeone Antonio 4 5
Brand Michael 3 6
Dollé Pascal 2 6
Rhinn Muriel 2 6 rhinn@igbmc.fr
1 Mouse Clinical Institute / Institut Clinique de la Souris , PHENOMIN, GIE CERBM , Illkirch Cedex , France
2 Institut de Génétique et de Biologie Moléculaire et Cellulaire, Centre National de la Recherche Scientifique, Institut National de la Santé et de la Recherche Médicale, Université de Strasbourg , Illkirch Cedex , France
3 DFG-Center for Regenerative Therapies / Cluster of Excellence, and Biotechnology Center, Technische Universität Dresden , Dresden , Germany
4 Institute of Genetics and Biophysics “Adriano Buzzati-Traverso” , Naples , Italy
5 IRCCS Neuromed , Pozzilli , Italy
Kubke M Fabiana
6 Senior authors.

Electronic publication date: 2013 Aug 29
Publication date: 2013
Volume: 1
Electronic Location ID: e142
Received 2012 Nov 27; Accepted 2013 Aug 4
Copyright: © 2013 Meziane et al.
Copyright year: 2013
Copyright holder: Meziane et al.
License: This is an open access article distributed under the terms of the Creative Commons Attribution License, which permits unrestricted use, distribution, and reproduction in any medium, provided the original author and source are credited.
License URL: https://creativecommons.org/licenses/by/3.0/

Keywords: Gbx genes, Spinal cord, Mouse mutant, Locomotion, GABAergic neurons

Funding: Agence Nationale de la Recherche ANR Neurosciences 2007 ANR Blanc 2011 Fondation pour la Recherche Médicale Equipe FRM 2007 Deutsche Forschungsgemeinschaft SFB-655 A3-Brand Association for Cancer Research (AIRC) Centre National de la Recherche Scientifique (CNRS) Institut National de la Santé et de la Recherche Médicale (INSERM) University of Strasbourg This work was supported by grants from the Agence Nationale de la Recherche (ANR Neurosciences 2007, ANR Blanc 2011), the Fondation pour la Recherche Médicale (Equipe FRM 2007), the Deutsche Forschungsgemeinschaft (SFB-655 A3-Brand), the Italian Association for Cancer Research (AIRC), and by institutional funding from the Centre National de la Recherche Scientifique (CNRS), Institut National de la Santé et de la Recherche Médicale (INSERM), and University of Strasbourg. Behavioral phenotyping was partly subsidized by the EUMODIC European Consortium and the Mouse Clinical Institute (MCI/ICS, Strasbourg). The funders had no role in study design, data collection and analysis, decision to publish, or preparation of the manuscript.

==============================
Dorsal horn neurons in the spinal cord integrate and relay sensory information to higher brain centers. These neurons are organized in specific laminae and different transcription factors are involved in their specification. The murine homeodomain Gbx1 protein is expressed in the mantle zone of the spinal cord at E12.5-13.5, correlating with the appearance of a discernable dorsal horn around E14 and eventually defining a narrow layer in the dorsal horn around perinatal stages. At postnatal stages, Gbx1 identifies a specific subpopulation of GABAergic neurons in the dorsal spinal cord. We have generated a loss of function mutation for Gbx1 and analyzed its consequences during spinal cord development. Gbx1−/− mice are viable and can reproduce as homozygous null mutants. However, the adult mutant mice display an altered gait during forward movement that specifically affects the hindlimbs. This abnormal gait was evaluated by a series of behavioral tests, indicating that locomotion is impaired, but not muscle strength or motor coordination. Molecular analysis showed that the development of the dorsal horn is not profoundly affected in Gbx1−/− mutant mice. However, analysis of terminal neuronal differentiation revealed that the proportion of GABAergic inhibitory interneurons in the superficial dorsal horn is diminished. Our study unveiled a role for Gbx1 in specifying a subset of GABAergic neurons in the dorsal horn of the spinal cord involved in the control of posterior limb movement.

Introduction

Perception of sensory inputs from both external and internal environments requires multiple levels of organization in the nervous system. The dorsal spinal cord plays critical roles in organizing responses to sensory input, and contains neurons that relay somatosensory information from sensory neurons in the periphery to motor neurons located in the ventral horns and to higher brain centers (for review: Helms & Johnson, 2003). These functions reside in a large number of distinct interneuron types that are arranged in an organized laminar structure in the dorsal horns (Rexed, 1952; Brown, 1981). Five parallel layers (laminae) have been defined in the murine spinal cord dorsal horn. These laminae are formed of unique combination of neurons, distinguished by their morphology and projections and by their gene expression profiles. The laminae receive different sensory input, with tactile perception mediated by myelinated axon bundles projecting to internal dorsal laminae (III, IV, V), and pain and temperature conveyed through unmyelinated axons that project to more superficial laminae (I, II) (for review: Caspary & Anderson, 2003). Proprioception is mediated by sensory neurons that project through the dorsal spinal cord to an intermediate zone which in turn projects to the ventral spinal cord where a direct connection is made with motoneurons (Brown, 1981; for review: Caspary & Anderson, 2003).

There are six early-born (in the mouse, by embryonic day E10-12.5) dorsal neuron populations called dI1-dI6 and two late-born (E11-E13) populations called dILA and dILB, defined by expression of specific homeodomain transcription factors (for review: Helms & Johnson, 2003; Lewis, 2006). These neurons can be further classified by their dependance on roof plate signaling for formation: class A (dI1-dI3) neurons depend on, whereas class B (dI4-dI6, dILA/B) neurons are independent of roof plate signals (Gross, Dottori & Goulding, 2002; Müller et al., 2002; for review: Helms & Johnson, 2003). The dorsal interneuron subtypes dI1-3 migrate ventrally, whereas a subset of dI4 and dI5 cells migrate laterally to populate the deep dorsal horn (laminae IV–V). The dILA/B subclasses migrate to the superficial dorsal laminae (I–III), and mediate pain and temperature sensitive circuits (for review: Caspary & Anderson, 2003).

The functional architecture of the mature dorsal horn is the result of developmental processes that involve cell-type specification and differentiation, as well as cell migration. Several events that control the specification of various neuronal subtypes in the spinal cord have been defined in recent studies (for reviews: Lee & Jessell, 1999; Briscoe & Ericson, 1999; Caspary & Anderson, 2003; Lewis, 2006). These studies demonstrate that homeodomain transcription factors play a central role during development of neurons in the dorsal horn (for reviews: Goulding et al., 2002; Helms & Johnson, 2003). Relatively few direct correlations have been made between dorsal interneuron progenitor classes and terminally differentiated cell types. However, formation of the proprioceptor pathway, which projects through the dorsal horn to the ventrally located motor neurons (Brown, 1981; Willis & Coggeshall, 1991) was shown to be dependent on Math1 (Bermingham et al., 2001; Gowan et al., 2001). Also, a dorsal horn-specific transcription factor, Drg11, is expressed in late born cells derived from dl5 precursors and is required for correct afferent fiber projections of nociceptive sensory neurons and correct dorsal horn morphogenesis (Chen et al., 2001; Rebelo et al., 2010). Finally, the Lbx1 gene is required for maturation of several dorsal horn cell types which later populate laminae I–III, and is critical for the correct projection of the nociceptive fibers into these laminae (Gross, Dottori & Goulding, 2002; Müller et al., 2002).

The gene encoding the homeodomain factor Gbx1 is expressed broadly in the mantle zone of the spinal cord at E12.5-13.5 (Rhinn et al., 2003; Waters, Wilson & Lewandoski, 2003; John, Wildner & Britsch, 2005). With the appearance of a discernable dorsal horn around E14, Gbx1 expression becomes more restricted, eventually defining a narrow layer in the dorsal horn around perinatal stages (John, Wildner & Britsch, 2005). Recently, immunohistological analysis showed that at E12.5, only a subpopulation of the Lbx1-positive cells coexpress Gbx1 (John, Wildner & Britsch, 2005). Lbx1 is a key determinant for the specification of class B neurons (Gross, Dottori & Goulding, 2002; Müller et al., 2002), suggesting that Gbx1-positive cells could correspond to class B neuron precursors (John, Wildner & Britsch, 2005). Late-born class B neurons comprise initially two neuron populations, dILA and dILB, which are born in an apparent salt and pepper pattern in the dorsal spinal cord. dILA neurons express Lbx1, Pax2, and Lhx1/5, whereas dILB cells express Lbx1, Lmx1b, and Tlx3 (Müller et al., 2002). At E12.5 and E14.5, Gbx1 neurons co-express the transcription factors Lhx1/5 and Pax2, but are negative for Lmx1b and Tlx3. This indicates that Gbx1 expression distinguishes a subpopulation of dILA neuronal cells (John, Wildner & Britsch, 2005). Furthermore, these authors show that GABA or Gad67 expressing neurons coexpress Gbx1, suggesting that Gbx1-positive cells may differentiate into GABAergic neurons.

To investigate the function of Gbx1 during dorsal horn development, we have generated mice bearing a mutation that ablates Gbx1 function. We report that Gbx1 knockout mice are viable and can reproduce as homozygous null mutants. However, the adult mutant mice display an altered gait during forward movement that specifically affects hindlimbs. This abnormal gait was evaluated by a series of behavioral tests, which revealed that locomotion is impaired, but not muscle strength or motor coordination. We then analyzed the development of the spinal cord dorsal horn in Gbx1−/− mice. Despite the clear behavioral phenotype, we did not observe changes in the expression of homeodomain factors regulating dorsal spinal cord development, suggesting that the development of the dorsal horn is not profoundly affected in Gbx1−/− mice. However, analysis of terminal neuronal differentiation revealed that expression of Gad67, a marker for GABAergic inhibitory interneurons, is diminished. Gbx1 is therefore required for the differentiation of inhibitory local circuit interneurons in the superficial dorsal horn, demonstrating a function for this transcription factor in the dorsal horn of the spinal cord.

Materials and Methods

Construction of a Gbx1 targeting vector

Genomic sequences encompassing the mouse Gbx1 gene were isolated from a 129SV genomic phage library, using a Gbx1 cDNA fragment previously characterized as a probe (Rhinn et al., 2003). A Gbx1 loss of function mutation was produced by homologous recombination in embryonic stem cells (Ramírez-Solis, Davis & Bradley, 1993). The targeting vector contained a 5.4 kb XmnI fragment (upstream arm), ending 33 bp upstream of the homeodomain sequence located in Gbx1 second exon, and a 1.6 kb KpnI fragment (downstream arm), whose sequence started 91 bp downstream from the homeodomain. These fragments were excised from the recombinant phage and cloned in the mutagenesis pGN vector (Le Mouellic, Lallemand & Brûlet, 1990) to generate the pGN-Gbx1 targeting vector (Fig. 1A). In this vector, the fragments are inserted on each side of a lacZ reporter gene and a neomycin resistance gene, and their insertion by homologous recombination in the Gbx1 gene will generate a 313 bp deletion encompassing the entire homeodomain (Fig. 1A).

Figure 1 Inactivation of the mouse Gbx1 gene by homologous recombination in embryonic stem (ES) cells.

(A) The upper drawing shows the restriction map of the wild-type locus, boxes and lines corresponding to exons and introns, respectively. The homeodomain sequence is in red. In the targeting vector (below), two Gbx1 genomic fragments (between the dashed lines) flank a lacZ reporter gene and the neomycin resistance gene (grey box), transcribed in the same orientation (thin arrow) as Gbx1. In the recombined locus (lower drawing), 313 bp of Gbx1 exon 2 (including the homeodomain) are replaced by the lacZ neo sequence. The location of the 3′ probe used for Southern blot analysis of ES cells is indicated in blue, and the PCR primers used to distinguish wild-type and recombined alleles for genotyping of animals (F1, R1, LacZ R2; see Materials and Methods) are also indicated. (B) Southern blot analysis of a targeted cell line ( + /−) in comparison to wild-type ( + / + ) HM-1 ES cells, using a probe external to the targeting vector 3′ homology arm. (C) Genotyping of wild-type ( + / + ), heterozygous ( + /−) or homozygous mutant (−/−) mice by PCR amplification of fragments specific for the wild-type (354 bp) or mutated allele (269 bp), using the F1, R1 and LacZ R2 primers. (D, E) Anti-Gbx1 immunostaining. At E18.5, Gbx1 protein is absent in the spinal cord of Gbx1−/− mice (E), compared to wild- type (D). Scale bars: 100 µm.

Transfection of embryonic stem cells and selection of targeted clones

HM-1 embryonic stem (ES) cells (Magin, McWhir & Melton, 1992) were cultured on neomycin-resistant mouse embryonic fibroblasts, as described in Robertson (1987). Ten µg of the pGN-Gbx1 targeting vector were linearized by digestion of the unique NotI restriction site, and electroporated into 2 × 107 ES cells resuspended in 750 µl HeBS medium (20 mM Hepes pH 7.05, 137 mM NaCl, 5 mM KCl, 0.7 mM Na2HPO4, 6 mM glucose), at 200 V, 960 µF. Positive selection was carried out for 11 days with 350 µg/ml G418. Resistant colonies were picked and DNA was extracted from a fraction (15) of the cells to perform Southern blot analysis to identify homologous recombination events. The probe used is an external fragment located immediately downstream to the targeting vector (Figs. 1A and 1B). Positive clones were expanded before freezing. The frequency of homologous recombination was 7 out of 350 clones analyzed.

Generation and genotyping of chimeric and mutant mice

After thawing, 10 to 15 ES cells were microinjected into blastocysts collected at E3.5 from C57BL/6 females mated with C57BL/6 males (for procedures: Nagy et al., 2003). Injected blastocysts were reimplanted in the uterine horn of pseudopregnant recipient females. Chimeric animals were back-crossed to C57BL/6J mice and germ-line transmission was scored by the presence of agouti coat pigmentation. Heterozygous offspring were identified by PCR genotyping. Tail tips were incubated in lysis buffer (50 mM Tris pH 8.0, 100 mM EDTA, 100 mM NaCl, 1% SDS, 0.6 mg/ml proteinase K) overnight at 55°C, phenol-chloroform extracted, ethanol precipitated and redissolved in 10 mM Tris–HCl, 1 mM EDTA pH 8.0 at a final concentration of 0.2–1.0 µg/µl. The presence of a wild-type or mutated allele was detected using three primers: a sense primer F1: 5′-GGTGACAGCGAGGACAGCTTCCT-3′, an antisense primer R1: 5′-CCCAGAACGACTGCTCACATTGC-3′, and an antisense primer LacZ R2: 5′-GGCCTCTTCGCTATTACGCCA-3′. The presence of a wild-type allele was detected using the F1/R1 primers which amplify a 354 bp fragment. The presence of a mutated allele was detected by using the F1/LacZ R2 primers which amplify a 269 bp fragment. Thirty cycles (denaturation: 1 min, 95°C, annealing : 1 min, 62° C; elongation : 30 s, 74°C) were performed, and the amplified products were separated by 2% agarose gel electrophoresis (Fig. 1C). Phenotypic and molecular analyses were performed after several generations of backcrosses (>5) to C57BL/6J mice, resulting in a nearly pure genetic background.

Tissue collection and sample preparation

Pregnant females obtained from natural matings (morning of vaginal plug was considered as E0.5) were sacrificed and fetuses were collected in phosphate-buffered saline (NaCl: 8.01 g/L, KCl: 0.2 g/L, Na2HPO4, 2H2O: 1.78 g/L, KH2PO4: 0.27 g/L, pH 7.5; hereafter abbreviated PBS 1×) after cesarean section. The specimens were dissected, fixed overnight in 4% paraformaldehyde (PFA) diluted in PBS 1×, pH 7.5, cryoprotected in 20% sucrose in PBS 1×, pH 7.5 and embedded in Shandon Cryomatrix (Thermo Electron Corporation) before freezing at −80°C. Cryosections (14 µm thickness, Leica CM3050S cryostat) sections were made in a coronal plane, collected on Superfrost slides, and stored at −80°C until use.

For whole-mount immunostaining or in situ hybridization, embryos were fixed overnight in 4% PFA, dried at room temperature, and stored at −20°C in 100% methanol.

In situ hybridization

In situ hybridization was performed with digoxigenin-labeled probes as previously described (Chotteau-Lelièvre, Dollé & Gofflot, 2006). Template DNAs were kindly provided by Drs K Jagla (Lbx1), C Birchmeier (Lmx1b), M Tessier-Lavigne (Netrin), AJ Tobin (Gad67) and P Gruss (Pax2), P Bouillet (Gbx2), B Giros (Slc17a6), F Chen (Drg11), R Krumlauf (Hoxb1), and F Rijli (Hoxa2). For all experiments 3 animals of each genotype, from 2 or more independent litters, were analyzed (except for Gbx2: Figs. S1A–S1D, n = 2). Cell countings were performed in the dorsal horn (Gad67, Pax2, Slc17a6) or ventral horn (Islet1) on 3 transverse sections for each animal, at comparable levels of the lumbar spinal cord (all sections were collected serially, with section planes being separated by 112 µm). Three animals of each genotype were thus analyzed for each marker.

All expression patterns were documented using a macroscope (Leica M420) or microscope (DM4000B, objective 10×), both connected to a Photometrics camera with the CoolSNAP (v. 1.2) imaging software (Roger Scientific, Chicago, IL). Cell counts were performed using the image J (NIH 1.45S) software. Blue labelled cells and unlabelled cells were counted manually with the cell counter plug-in. Three sections separated by 112 µm in 3 independent embryos were counted for each condition, and statistical significance of cell counts was validated with a two way measures analysis of variance (ANOVA) with the first variable as fixed effect (i.e., genotype) and a second variable as random effect of repeat observations on the same individual. The level of significance was set at p < 0.05. Graphs represent averages of counting values and SEM.

Immunohistochemistry

After antigen unmasking in citrate buffer 0.01 M (pH 6) during 15 min in a microwave oven, sections were treated in H2O2 3% in PBS 1×, pH 7.5 for 5 min, rinsed in PBS 1×, then blocked in PBS 1×, pH 7.5 containing 0.25% Triton-X100, 5% normal goat serum and incubated overnight at 4°C with rabbit anti-Gbx1 (kindly provided by Dr. S Britsch; 1:500), rabbit anti-calbindin D-28K (Chemicon, 1:1000), rabbit anti-Peripherin (Chemicon, 1:500), or mouse anti-Islet1 (40.2D6, concentrated, Developmental Studies Hybridoma Bank, Iowa City, IA, 1:100) in PBS 1×, pH 7.5 containing 0.25% Triton-X100, 5% normal goat serum followed by species-specific biotin-coupled secondary antibodies (1:400, Jackson Laboratories) diluted in PBS 1×, pH 7.5. Detection was performed using a Vectastain Elite ABC Kit, following the manufacturer’s instructions. Nissl staining was performed by incubation in 0.5% cresyl violet in water for 15 min. TUNEL was performed using the APOPTAG® Peroxidase In Situ Apoptosis detection kit (Millipore). For all experiments 3 animals of each genotype were analyzed.

Behavioral phenotyping procedures

Cohorts of 10-week-old male and female Gbx1−/− mice in a C57BL/6J genetic background (7 males and 8 females), with their wild-type (WT, 10 males and 9 females) counterparts, were used in this study. Mice were group housed and allowed 2 weeks acclimation in the phenotyping area with controlled temperature (21–22°C) under a 12–12 h light-dark cycle (lights on 7 am–7 pm), with food and water available ad libitum. Testing started at 10 weeks of age, and all procedures were carried out in accordance with European institutional guidelines. Behavioral tests were performed successively for each cohort of mice, during the light phase of the circadian cycle, according to a pipeline established by the European Mouse Disease Clinic (EUMODIC pipeline 2), by trained experimenters familiar with observation of normal gait patterns in mice. Detailed procedures for each test are available at the URL: http://www.empress.har.mrc.ac.uk/viewempress/index.php?pipeline=EUMODIC+Pipeline+2.

Neurological examination

General health and basic sensory motor functions were evaluated using a modified SHIRPA protocol (Brown, Chambon & Hrabé de Angelis, 2005; protocol at http://www.empress.har.mrc.ac.uk/viewempress/index.php?pipelineprocedure=EUMODIC+Pipeline+2~Modified+SHIRPA). This analysis is adapted from the procedure developed by Irwin (1968) and from the SHIRPA protocol (Hatcher et al., 2001). It provides an overview of physical appearance, body weight, neurological reflexes and sensory abilities.

Rotarod test

This test evaluates motor coordination and balance by measuring the ability of animals to maintain balance on a rotating rod (Bioseb, Chaville, France). Mice were given three testing trials during which the rotation speed accelerated from 4 to 40 rounds per min (rpm) over 5 min. Trials were separated by 5–10 min intervals. The average latency (time to fall from the rotating rod) of the three trials was used as the index of motor coordination performance.

Grip test

This test measures the maximal muscle strength (g) using an isometric dynamometer connected to a grid (Bioseb). Mice were allowed to grip the grid either with the forepaws or with both the forepaws and hindpaws, then were pulled backwards until they released the grid. Each mouse was submitted to 3 consecutive trials immediately after the modified SHIRPA procedure. The maximal strength developed by the mouse before releasing the grid was recorded and the average value of the three trials was adjusted to body weight.

Beam walking

This test is used to evaluate fine motor coordination and proprioceptive function. The apparatus used were a 2 cm diameter and 110 cm long wooden beam, elevated 50 cm above the ground. A goal box (12 × 12 × 14 cm) is attached at one extremity of the beam. Animals were first habituated to the goal box for 1 min. They were then submitted to 3 training trials during which they were placed at different points of the beam, with the head directed to the goal box, and allowed to walk the corresponding distance to enter the goal box.

After training, animals were submitted to 3 testing trials during which they were placed at the extremity of the beam opposite to the goal box and allowed to walk the beam distance and enter the goal box. The latency to enter the goal box and the number of slips (when one or both hindpaws slipped laterally from the beam) were measured.

Hot plate test

The mice were placed into a glass cylinder on a hot plate (Bioseb) adjusted to 52°C, and the latency of the first pain reaction of any hindlimb (licking, flinches) was recorded, with a maximum of 30 s testing.

Electrophysiological measurements

Electrophysiological recordings were performed under ketamine-xylazine anesthesia (100 and 10 mg/kg body weight, respectively) using a Key Point electromyograph apparatus (Medtronic, France). Disposal scalp needle electrodes were used (ref 9013R0312, Medtronic). The body temperature was maintained at 37°C with a homeothermic blanket (Harvard, Paris, France). For measuring the sensory nerve conduction velocity (SNCV), recording electrodes were inserted at the proximal part of the tail and stimulating electrodes placed 20 mm from the recording needles towards the extremity of the tail. A ground needle electrode was inserted between the stimulating and recording electrodes. Caudal nerve was stimulated with a series of 20 pulses of 0.2 ms duration at a supramaximal intensity of 8 mA. The average response is included for statistical analysis. The compound muscle action potential (CMAP) was measured in gastrocnemius muscle after sciatic nerve stimulation. For this purpose, stimulating electrodes were placed at the level of the sciatic nerve at 1 cm from the vertebral column, and recording electrodes placed in the gastrocnemius muscle. A ground needle was inserted in the contralateral paw. The sciatic nerve was stimulated with a single 0.2 ms pulse at a supramaximal intensity of 8 mA. The amplitude (mV) and the distal latency of the responses (ms) were measured.

Anxiety-related behavior — open field test

Mice were tested in automated open fields (Panlab, Barcelona, Spain), each virtually divided into central and peripheral regions. The open fields were placed in a room homogeneously illuminated at 150 lx. Each mouse was placed in the periphery of the open field and allowed to explore the apparatus freely for 20 min, with the experimenter out of the animal’s sight. The distance traveled, the number of rears, and time spent in the central and peripheral regions were recorded over the test session. The latency and number of crosses into as well as the percent time spent in center area are used as an index of emotionality/anxiety.

Sensorimotor gating — auditory startle reflex reactivity and pre-pulse inhibition (PPI)

Acoustic startle reactivity and pre-pulse inhibition of startle were assessed in a single session using standard startle chambers (SR-Lab Startle Response System; San Diego Instruments). Ten different trial types were used: acoustic startle pulse alone (110 db), eight different prepulse trials in which either 70, 75, 85 or 90 dB stimuli were presented alone or preceding the pulse, and finally one trial (NOSTIM) in which only the background noise (65 dB) was presented to measure the baseline movement in the Plexiglas cylinder. In the startle pulse or prepulse alone trials, the startle reactivity was analyzed, and in the prepulse plus startle trials the amount of PPI was measured and expressed as percentage of the basal startle response.

Statistical analyses

Data were analyzed using unpaired Student t-test, one way or repeated measures analysis of variance (ANOVA) with one between factor (genotype) and one within factor (time). Qualitative parameters (i.e., some of the clinical observations) were analyzed using χ2 test. The level of significance was set at p < 0.05.

Animal ethics statement

Animal experimentation protocols were reviewed and approved by the Direction Départementale des Services Vétérinaires (agreement #67-172 to HM, 67-189 to PD, and institutional agreement #D67-218-5 for animal housing) and conformed to the NIH and European Union guidelines, provisions of the Guide for the Care and Use of Laboratory Animals, and the Animal Welfare Act.

Results and Discussion

Gbx1-deficient mice are viable, but display a typical duck-like gait

A loss of function allele for the Gbx1 gene was generated by homologous recombination in murine embryonic stem cells (see Materials and Methods). The mutated Gbx1 allele is devoid of the entire homeodomain-coding sequence and ∼100 adjacent nucleotides (Figs. 1A–1C). After generation of germ-line transmitting chimeras, heterozygous mutant mice (Gbx1+/−) were found to be viable, fertile and apparently normal. After intercrossing Gbx1+/− mice, Gbx1−/− mutants (generated in a C57BL/6J genetic background) were born in the expected Mendelian ratio. Immunohistochemistry performed with an anti-Gbx1 antibody confirmed the absence of detectable Gbx1 protein in the spinal cord of E18.5 Gbx1−/− mutants (Figs. 1D and 1E). We also checked the expression of Gbx2 from E12.5 to E18.5 (Fig. S1) to exclude a potential compensatory expression due to the loss of function of Gbx1. A subtle increase of Gbx2 mRNA expression might occur in spinal cord cells of Gbx1−/− mice at E12.5-14.5, however, this increase was no longer detected at E16.5 or E18.5 (Fig. S1). This subtle Gbx2 increase could partially compensate for the loss of Gbx1, leading to the absence of phenotypic abnormalities at early developmental stages. Interestingly, when observed by 10 weeks of age, most mutants displayed a typical, unevenness in walking (“duck-like”) gait (Fig. 2 and Video S1). Both male and female Gbx1−/− mice were fertile and had a normal life span.

Figure 2 Abnormal phenotype of a 10 week-old Gbx1−/− mouse when walking.

Sequential pictures compare the normal gait of a wild-type mouse (A) and the abnormal gait (“duck-like” walk) of a Gbx1−/− mutant when walking. (B) A movie of these mice is available (Video S1).

General health and sensorimotor abilities in adult Gbx1 mutants

Gbx1−/− males and females had a normal body weight (Table 1) and a normal overall physical appearance. However, many of the Gbx1−/− mutants showed significantly abnormal gait (χ2 ≥ 5.20, p < 0.05). Indeed, 43% of Gbx1 mutant males and 63% of Gbx1 mutant females displayed lack of fluidity in movement, and limping related to hyper-flexion followed by hyper-extension of one or both hindpaws (Table 1; Fig. 2; Video S1). Gbx1−/− males and females also showed significantly reduced short-term locomotor activity following immediate transfer for the modified SHIRPA test, as compared to WT counterparts (t ≥ 3.46, p < 0.01) (Table 1). The other features of general health and basic neurological reflexes were not affected in Gbx1 mutants.

Table 1 Effects of Gbx1 mutation on body weight, basic neurological reflexes, specific motor abilities and pain sensitivity.

Mice were analyzed at 10 weeks of age. Statistically different parameters in wild-type vs mutants appear in bold.

		Males	Females	
		Wild-type	Gbx1 −/−	Wild-type	Gbx1 −/−	
Body weight (g)		26.57 ± 0.73	25.40 ± 0.48	19.81 ± 0.32	20.88 ± 0.89	
Gait (% abnormal)		0	43% **	0	63% **	
Short-term locomotor activity
(number of squares crossed in 30 s)		26.70 ± 1.37	19.71 ± 1.39 **	28.33 ± 1.00	22.00 ± 1.57 **	
Rotarod-4 to 40 rpm in 5 min (s)		123.87 ± 17.56	95.11 ± 15.08	116.33 ± 9.10	130.25 ± 14.99	
Grip strength (adjusted to body weight) (g)	2 paws	3.97 ± 0.18	4.00 ± 0.24	3.75 ± 0.17	3.38 ± 0.26	
	4 paws	8.30 ± 0.24	7.72 ± 0.37	7.00 ± 0.24	7.22 ± 0.54	
Hot plate (s)		13.43 ± 1.26	17.13 ± 1.08 *	12.72 ± 1.15	15.03 ± 1.63	
Notes.

* p < 0.05.

** p < 0.01 vs wild-type; Student t-test.

When tested for specific motor abilities, motor coordination performance measured in the rotarod test (t ≤ 1.29, NS) and the muscle strength (grid grip) test (t ≤ 1.38, NS) were not affected in Gbx1−/− males and females (Table 1). In the beam walking test, the latency to cross the beam was increased (t15 = 3.71, p < 0.01 for females; non significant for males) and the number of slips was slightly increased (even if not significantly), especially in Gbx1−/− females (Fig. 3). In the open field test, there was a significant effect of genotype concerning locomotor activity [F(1, 30) = 6.51, p < 0.05], reflecting reduced locomotion in all Gbx1−/− animals. When considering each gender separately, both Gbx1−/− males and females tended to have reduced locomotor activity over the testing period (although not statistically significant, p = 0.09) (Fig. 4). The average speed during motion was also significantly lower in Gbx1−/− males and females than in WT (t ≥ 3.36, p < 0.01) (Fig. 4). The number of entries into, and the percentage of time spent in, the center of the arena also differed between genotypes [F(1, 30) ≥ 14.48, p < 0.001]. Both Gbx1−/− males and females had significantly decreased number of entries and spent less time in the center of the open field than WT counterparts (t ≥ 2.62, p < 0.05) (Fig. 4), which might reflect increased anxiety in Gbx1−/− mutants. The reduced exploration of the center might also be due to the observed reduced locomotor activity of Gbx1−/− mutants. Altogether, these data show that Gbx1−/− mutant mice have a clear defect in locomotion, although this defect does not appear to result from a coordination problem or a muscle strength deficiency.

Figure 3 Effects of Gbx1 mutation on the latency and number of slips in the beam walking test.

∗∗p < 0.01 vs WT; Student t-test.

Figure 4 Open field performance of wild-type (WT) and Gbx1−/− mice.

The distance traveled over the 20 min period of test reflects locomotor activity. The average speed was calculated during movement in the whole arena for the entire period of testing. Exploration of the central part of the open fied is expressed as the number of entries and percentage of time spent in the center. ∗p < 0.05 and ∗∗p < 0.01 vs WT; Student t-test.

To test whether ablation of Gbx1 could affect sensory response, we measured the response of Gbx1 mutant mice in a hot plate test. The withdrawal latency was higher in Gbx1−/− males (but not in females) than in WT (t15 = 2.10, p = 0.05) (Table 1), suggesting reduced thermal pain sensitivity in Gbx1 mutant males.

The consequence of Gbx1 inactivation on acoustic startle and pre-pulse inhibition of startle reflex was also evaluated. Regardless of gender, the startle reactivity was comparable between WT and Gbx1−/− mice for all the acoustic stimuli including the startling pulse [Genotype F(1, 30) ≤ 0.83, Sex F(1, 30) ≤ 0.85, Genotype*Sex F(1, 30) ≤ 1.11, NS] (Fig. 5). When the startling pulse was preceded with prepulses with lower intensities, the PPI level was also comparable between genotypes [Genotype F(1, 30) = 0.55, Sex F(1, 30) = 0.32, Genotype*Sex F(1, 30) = 0.11, NS)] (Fig. 5). Furthermore, electromyography (EMG) measurements revealed that the sensory nerve conduction velocity (SNCV) differed significantly between genotypes [F(1, 29) = 7.31, p < 0.05]; indeed, Gbx1−/− females had significantly increased SNCV (t14 = 2.83, p < 0.05) (Table 2), as measured at the level of the caudal nerve. On the other hand, the latency and amplitude of the gastrocnemius muscle response evoked by sciatic nerve stimulation were comparable between genotypes [F(1, 29) = 1.63, NS].

Figure 5 Startle reactivity and pre-pulse inhibition in wild-type (WT) and Gbx1−/− mice.

Startle reactivity to background noise (65 dB), or to 70, 80, 85, 90 dB acoustic stimulation, and startle reflex to a 110 dB stimulus are presented. The percentage of pre-pulse inhibition of the startle response is displayed as a percentage of the pre-pulse intensity. WN, white noise; P, acoustic pulse intensity; ST, acoustic startle to 110 dB; PP, pre-pulse intensity.

Table 2 Effects of Gbx1 mutation on sensory nerve conduction velocity.

The sensory nerve conduction velocity was measured at the level of the caudal nerve. The latency and the amplitude of gastrocnemius muscle response evoked by sciatic nerve stimulation were also recorded.

		Males	Females	
		Wild-type	Gbx1 −/−	Wild-type	Gbx1 −/−	
Sensory nerve conduction velocity (m/s)		70.33 ± 1.70	72.20 ± 1.53	63.93 ± 2.35	71.81 ± 1.50 *	
Gastrocnemius M-wave	Latency (ms)	0.93 ± 0.06	0.91 ± 0.06	0.99 ± 0.08	0.84 ± 0.05	
	Amplitude (mV)	43.99 ± 3.15	44.60 ± 5.87	46.46 ± 6.70	55.41 ± 6.03	
Notes.

* p < 0.05 vs wild-type; Student t-test.

In summary, we used a variety of behavioral and electrophysiological phenotyping tests to evaluate sensory and motor functions in Gbx1 mutant mice. Decreased exploratory behavior was found in the open field test and following immediate transfer during clinical observations. Exploration of the central part of the open field arena was significantly decreased in Gbx1−/− males and females which might suggest increased anxiety in these mutants. However, this could also be due to the reduced locomotor activity of the mutants. Indeed, Gbx1−/− mice also showed decreased average speed with no significant effect on the distance travelled in the open field. Their altered gait during forward movement might explain the reduced speed and locomotor activity in the open field, which could not be attributed to defects in motor coordination or muscle strength. Abnormal gait may suggest proprioceptive-like deficits as indicated by abnormal performance in beam walking, the test used for evaluation of proprioceptive functions, which was statistically significant only for Gbx1−/− females. Although no direct link can be clearly identified between motor deficits and sensory functions, we cannot exclude mutual interdependence of abnormal gait and sensory deficits indicated by reduced responses in hot plate test and increased SNCV, which were penetrant to a different degree in null-mutant males and females.

Gbx1−/− mice do not show obvious hindbrain patterning defects

Gbx genes are related to the Drosophila unplugged gene, which acts during development of the tracheal system, and for specification of neuroblast sublineages (Chiang, Young & Beachy, 1995; Cui & Doe, 1995). There are two Gbx genes in amniote species (human, mouse and chicken), as well as in zebrafish (Lin et al., 1996; Bouillet et al., 1995; Shamim & Mason, 1998; Niss & Leutz, 1998; Rhinn et al., 2003). Previous studies showed that in mice, Gbx2 is involved in early specification of the midbrain-hindbrain boundary (MHB) organizer, a signaling center that will pattern the anterior hindbrain rhombomeres (Wassarman et al., 1997; Waters & Lewandoski, 2006; for review: Rhinn & Brand, 2001; Simeone, 2000). In zebrafish it was shown that Gbx1 acts during early positioning of the MHB, whereas Gbx2 functions at later stages, once the MHB is established (Rhinn et al., 2003; Rhinn et al., 2009; Burroughs-Garcia et al., 2011). In mice, Gbx1 is not expressed at the MHB as is the case during early zebrafish development. Its expression starts at E7.75 in the prospective hindbrain, spanning rhombomeres 2 to 7 during the segmentation phase (Rhinn et al., 2003; Waters, Wilson & Lewandoski, 2003). This suggested that Gbx1 might be involved in early embryonic hindbrain patterning, which could underlie behavioral deficits associated with loss of Gbx1 function. To assess for possible rhombomeric abnormalities in Gbx1−/− mutants, we performed whole-mount in situ hybridizations at E9.5 with several markers, including Hoxb1 and Hoxa2. This analysis did not show any molecular or structural abnormality of the hindbrain rhombomeres in Gbx1−/− embryos (Fig. S2). This suggests that Gbx1 is not required for early hindbrain patterning, in contrast to its mouse homologue Gbx2 (Wassarman et al., 1997; Waters & Lewandoski, 2006). Analysis of hindbrain derivatives (brain stem and cerebellum) at E18.5 using Gad67 as a differentiation marker also did not reveal any difference in Gbx1−/− versus wild-type mice (Fig. S3).

Development of the spinal cord dorsal horn in Gbx1 mutant mice

At E12.5, the expression domains of Gbx1 and Gbx2 overlap, both being expressed in the ventricular and mantle zones of the dorsal spinal cord (Rhinn et al., 2003; Waters, Wilson & Lewandoski, 2003). As Gbx2 expression is downregulated after E12.5, both genes are only transiently coexpressed in dorsal spinal cord progenitor cells, and Gbx1 is the only Gbx gene persistently expressed during later dorsal horn development (John, Wildner & Britsch, 2005).

Thus, the prominent expression of Gbx1 in the dorsal horn could be relevant for the abnormal gait phenotype of adult Gbx1 mutant mice, which led us to ask whether Gbx1 is required for the maturation and/or specification of neurons of the dorsal horn during development. Nissl staining of E18.5 spinal cord sections revealed no obvious difference between the dorsal horn of wild-type and Gbx1−/− animals at thoraco-lumbar levels (Figs. 6A and 6B). Despite the clear behavioral phenotype, we were unable to identify any consistent alteration in the expression of several molecular markers of dorsal spinal cord cell populations in Gbx1−/− embryos. These markers included the genes encoding the transcription factors Lbx1 (Gross, Dottori & Goulding, 2002; Müller et al., 2002) (Figs. 6C and 6D), Lmx1b (Chen et al., 2001) (Figs. 6E and 6F) and the axon guidance molecule Netrin-1 (Leonardo et al., 1997) (Figs. 6G and 6H) analyzed at E12.5, 14.5, 16.5 and 18.5, and shown (Fig. 6) at E16.5.

Figure 6 Absence of morphological and molecular abnormalities in the developing dorsal horn of Gbx1−/− mice.

Sections through the spinal cord of wild-type (A, C, E, G) and Gbx1−/− (B, D, F, H) mice at E18.5 (A, B) and E16.5 (C–H) are shown. All sections are at the lumbar level. (A, B) Nissl-stained sections. No differences are detectable between wild-type and mutants (n = 3). In situ hybridizations for two transcription factor encoding genes, Lbx1 (C, D) and Lmx1b (E, F), and for the axon guidance molecule netrin-1 (G, H), are shown (n = 3 for each marker). No differences are observed between wild-type and mutants. Scale bars: 100 µm.

Projection pattern of primary sensory afferents in the dorsal horn of Gbx1−/− mutants

We examined the development of primary sensory afferent projections to the dorsal horn, which are well defined at E18.5, in Gbx1 mutant mice. The projections of cutaneous nociceptive sensory neurons begin to invade the spinal grey matter by E12.5 (Ozaki & Snider, 1997). Immunostaining with an anti-calbindin-28K antibody at E16.5 and E18.5 marks a subset of cutaneous neurons and their afferent fibers (Honda, 1995; Chen et al., 2001). By E18.5, calbindin + fibers have invaded the dorsal horn of wild-type and Gbx1 mutants (Figs. 7A–7B′). The Drg11 gene is required for the projection of cutaneous sensory afferent fibers to the dorsal spinal cord (Chen et al., 2001). In Gbx1−/− mutant mice, Drg11 expression was not affected in the dorsal horn at E12.5, 14.5, 16.5 or 18.5 (Figs. 7C and 7D and data not shown). Altogether, these data suggest that there are no defects in patterning of sensory afferent fiber projections to the dorsal horn, which selectively affects cutaneous afferents, although the markers used cannot rule out other types of patterning differences (for instance from primary afferents that do not label with calbindin).

Figure 7 Developmental progression of afferent projections in the dorsal horn of Gbx1−/− mice.

(A–B′) Anti-calbindin-D28K antibody staining. At E18.5, calbindin fibers have already entered the spinal gray matter in wild-type (A, A′) and Gbx1−/− specimens (B, B′; n = 3). Panels A′, B′ are higher magnifications of the areas boxed in A, B. (C, D) Expression of Drg11 in wild-type at E18.5 (C) and Gbx1 mutant (D) mice. Mutant specimens were indistinguishable from wild-types (n = 3). (E–H) Anti-peripherin antibody staining at E18.5 (E, F) and E16.5 (G, H). This staining reveals similar ingrowth of group IA muscle sensory afferents that grow to the ventral spinal cord (arrows) in wild-type (E, G) and mutant (F, H) (n = 3 for each stage). Scale bars: 100 µm (A′, B′: 50 µm).

We further examined proprioceptive afferents at E18.5 by using antibodies to peripherin (Escurat et al., 1990). No consistent difference between wild-type and Gbx1−/− mice was observed at the level of proprioceptive fibers that extend toward motoneurons and interneurons in the deep dorsal horn, or at the level of fibers that enter into the spinal gray matter, at E18.5 (Figs. 7E and 7F) or E16.5 (Figs. 7G and 7H). During the revision of our manuscript, another Gbx1 mutant allele was described (Buckley et al., 2013). In contrast to our observations and at a comparable stage, those mutants show disorganized peripherin expression, together with a decrease of Islet1-expressing cells in the ventral horn of the lumbar spinal cord (Buckley et al., 2013). This led us to analyze Islet1-expressing cells in ventral motor neurons in our Gbx1 mutants. No differences in the number of Islet1 + cells within the lumbar ventral spinal cord were found at E14.5, E16.5 (Fig. S4) or E18.5 (data not shown). Thus, in contrast to the data of Buckley et al., our analysis does not suggest a defect in the assembly of the proprioceptive sensorimotor circuit. As the same Gbx1 exon (exon 2) was targeted in both loss of function alleles, the reason for the phenotypic discrepancy remains unclear, although it should be mentioned that the mice might have different genetic backgrounds.

Reduced GABAergic neuronal differentiation in Gbx1−/− mutants

Gbx1 is first expressed in the ventricular zone of the spinal cord at E11.5 (Rhinn et al., 2003; Waters, Wilson & Lewandoski, 2003). Then at E12.5-E13.5, it is broadly expressed in the mantle zone of the dorsal spinal cord. At E14 with the appearance of a distinguishable dorsal horn, Gbx1 expression becomes more restricted. At E12.5, Gbx1 is coexpressed with Lbx1; thus Gbx1 cells correspond to class B neurons (John, Wildner & Britsch, 2005). As described in the introduction, late-born class B neurons comprise initially two populations, dILA and dILB. Because Gbx1 neurons co-express Lhx1/5 and Pax2, but not Lmx1b and Tlx3, it has been suggested that these neurons correspond to the dILA neuronal subtype (John, Wildner & Britsch, 2005). It has been shown that dILA neurons undergo GABAergic differentiation (Cheng et al., 2004), and as mature GABA + neurons they continue expressing Gbx1 (John, Wildner & Britsch, 2005). We therefore analyzed GABAergic neurons in the spinal cord of Gbx1−/− mutants, which we identified by expression of glutamic acid decarboxylase GAD67, an enzyme that regulates GABA synthesis. At E18.5, Gad67-expressing cells are found throughout the developing spinal cord of control mice (Somogyi et al., 1995). Importantly, Gad67 expression was reduced in the dorsal spinal cord of Gbx1 mutant mice (Figs. 8A–8D), i.e., there was a 16% decrease in the proportion of Gad67-expressing cells (Fig. 8I). This may reflect an abnormal development of GABAergic neurons, which in consequence could lead to abnormal control of neuronal network in dorsal horn, possibly affecting inhibitory circuits throughout the spinal cord. This finding was strengthened by analysis of Pax2, another gene expressed in GABAergic cells in the spinal cord (Cheng et al., 2004), with cell countings corroborating the decrease in the proportion of GABAergic cells (Figs. 8E, 8F and 8I). Furthermore, it is known that during early post-natal period, the GABA pathway switches from excitatory to inhibitory in mouse (Ben-Ari et al., 2007). This shift was shown to occur within the first two weeks of age in hippocampal and spinal motor neurons in mouse (Stein et al., 2004) as well as in lamina I in rats (Sibilla & Ballerini, 2009). Also, it was mentioned that the switch depends on the species, sex, brain structures and neuronal types (Ben-Ari et al., 2007) and it was shown using a model system of cultured hippocampal neurons that the switch is triggered by GABAergic activity itself (Ganguly et al., 2001). Interestingly, when Gbx1−/− pups were checked visually around weaning every morning (analysis done on 5 litters, 42 pups, 8 Gbx1−/− mutants), the locomotor deficits were first observed around post-natal days (P)16-17. Taking into account an eventual delay due to diminished GABA activity, the appearance of the locomotion defect could coincide with the time point at which the GABA pathway switches from excitatory to inhibitory.

Figure 8 Abnormal GABAergic differentiation in Gbx1−/− mice.

Expression of Gad67 in wild-type (A, C) and Gbx1−/− (B, D) mice at E18.5 (n = 3). Higher magnification views (C, D; areas boxed in A, B) show the dorsal horn, in which cell countings were performed. Expression of Pax2 in wild-type (E) and Gbx1−/− (F) mice at E18.5 (n = 3). Expression of Slc17a6 in wild-type (G) and Gbx1−/− (H) mice at E18.5 (n = 3). (I) Countings (percentages of labelled vs total cells) revealed that the proportion of Gad67 + cells is diminished by 16% in Gbx1−/− mice (50.89% ± 2.61 Gad67 + cells in WT; 34.85% ± 1.84 in Gbx1−/− mice; Genotype F(1, 4) = 223.85, ∗∗∗p < 0.001, Sections F(2, 8) = 2.22, NS, Genotype* Sections F(2, 8) = 1.23, NS). Also, the proportion of Pax2 + cells is diminished by 14.7% in Gbx1−/− mice (58.57% ± 4,03 Pax2 + cells in WT; 42.41% ± 5.96 in Gbx1−/− mice; Genotype F(1, 4) = 449.36, ∗∗∗p < 0.001, Sections F(2, 8) = 3.34, NS, Genotype∗ Sections F(2, 8) = 6.3, p < 0.05). In contrast, countings revealed that the proportion of Slc17a6 + cells is increased by 14.4% in Gbx1−/− mice (50.96% ± 1.84 Slc17a6 + cells in WT; 65.16% ± 2.94 in Gbx1−/− mice; Genotype F(1, 4) = 688.84, ∗∗∗p < 0.001, Sections F(2, 8) = 1.004, NS, Genotype*Section F(2, 8) = 4.73, p<0.05). Scale bars: 100 µm.

We also observed that Gad67 expression was unchanged in the brain stem and cerebellum of E18.5 Gbx1−/− mutants (Fig. S3), arguing against an involvement of these structures in the observed phenotype.

We next addressed the question of whether the observed decrease of GABAergic cells is due to neuronal cell death or to a possible change of GABAergic to glutamatergic fate. TUNEL experiments were performed at various stages (E12.5, 14.5, 16.5, 18.5; Fig. 9 and data not shown). As expected, natural cell death occurs mainly in the developing spinal ganglia and ventral spinal cord (Figs. 9A and 9B; White et al., 1998) and natural cell death is suggested to be over by E15.5 (Figs. 9C and 9D; White et al., 1998). Our analysis showed no abnormal apoptosis in the dorsal spinal cord of Gbx1−/− mice (Figs. 9B and 9D). This finding would exclude the decrease of Gad67-expressing cells due to cell death, and suggest that Gbx1 is not required for cell survival. We then analyzed expression of Slc17a6, encoding VGLUT2, a vesicular glutamate transporter expressed in glutamatergic neurons (Kaneko & Fujiyama, 2002). At E18.5, Slc17a6-expressing cells were increased in the dorsal spinal cord of Gbx1 mutant mice (Figs. 8G and 8H). This finding suggests that part of the “missing” GABAergic cells may have differentiated into glutamatergic neurons.

Figure 9 Examples of TUNEL labeling of lumbar spinal cord sections of wild-type (A, C) and Gbx1−/− (B, D) mice.

Sections are shown at E12.5 (A, B) (n = 3) and E18.5 (C, D) (n = 3). Some TUNEL-labelled cells are seen in the dorsal root ganglia (drg) and in the ventral spinal cord (magnified in upper insets) at E12.5, in both wild-type and mutant. (E) Apoptotic cells in the interdigital mesenchyme of an E13.5 forelimb are shown as a positive control. Scale bars: 100 µm.

Glutamate and GABA are the main neurotransmitters for excitatory and inhibitory neurons, respectively, in the vertebrate brain. These neurotransmitters are usually expressed in a mutually exclusive manner (Bellocchio et al., 2000; Fremeau et al., 2001). In the dorsal horn of the spinal cord, most ascending projection neurons and a subset of local circuit interneurons are excitatory and are glutamatergic. These neurons are modulated by local inhibitory neurons, many of which are GABAergic (for reviews: Melzack & Wall, 1965; Malcangio & Bowery, 1996; Dickenson, 2002). Thus, GABA may inhibit transmitter release from primary afferent fibers. The output neurons of the dorsal horn are projection neurons, relaying sensory information to several brain areas. However, the majority of dorsal horn neurons are local circuit interneurons that do not project outside of the spinal cord. The output of projection neurons is influenced by local excitatory and inhibitory neurons (Todd, 2010; Larsson & Broman, 2011; Guo et al., 2012). In Gbx1 mutants, the reduction in the proportion of GABAergic neurons, and the possible switch of some of these neurons to a glutamatergic identity, may disrupt neuronal circuitry, becoming phenotypically apparent at adult stages as measured by abnormal performance in several behavioral tests. Further electrophysiological studies will be necessary to link the decrease of GABAergic neurons to the abnormal gait observed in adult Gbx1 mutant mice.

Conclusion

We have generated Gbx1−/− loss of function mutant mice, and investigated the development of the spinal cord dorsal horn in these mutants. Gbx1−/− mutants are viable and fertile, but display an altered gait during forward movement that specifically affects hindlimbs, beginning at post-natal days 16-17. This abnormal gait, documented by a series of behavioral tests, is not due to deficits in muscle strength or motor coordination. Although reduced performance of Gbx1−/− mice in beam walking, a test used in studies of proprioception, could potentially suggest proprioceptive deficits, such a hypothesis is not fully supported by at least two observations: (i) the incomplete penetrance of this phenotype because significant deficits were observed only in females, and (ii) by molecular data, which did not reveal deficits in the assembly of proprioceptive sensory afferents in the ventral or intermediate zone, described previously (Brown, 1981) as their target regions before contacting motoneurons.

Some of the deficits, such as altered sensory nerve conduction velocity, are significantly altered in females, whereas significant difference in hot plate performance was identified only in males. Although such differences could reflect sexual dimorphism, it is difficult to draw such a conclusion as definitive for two major reasons: (i) in some tests where a significant difference in performance was observed for one gender, the opposite gender may display a similar tendency, which remained non statistically significant; (ii) if, for example, females would be more prone to effects of Gbx1 mutation we could expect to find them less performant in different tests; however, the gender effects were inconsistent and concerned males or females depending on the measured parameter.

The spinal cord dorsal horn largely consists of inhibitory (GABAergic) and excitatory (glutamatergic) neurons that modulate somatosensory inputs from the periphery, including pain, temperature and mechanoreception (Glasgow et al., 2005). Our analysis of major neuronal classes revealed a reduced proportion of GABAergic inhibitory interneurons expressing Gad67 in the superficial dorsal horn of Gbx1−/− mice. Gbx1 may therefore be functionally required for the differentiation of local inhibitory interneurons in the dorsal horn, corroborating a previous report of Gbx1 expression in a specific subset of GABAergic neurons in this region of the spinal cord (John, Wildner & Britsch, 2005). Furthermore, our findings suggest that Gbx1 functions as a gene that promotes GABAergic over glutamatergic differentiation in the dorsal horn. A disruption in the balance between inhibitory and excitatory neuronal activity could explain the phenotype observed in Gbx1 mutants. Indeed, the imbalance of inhibitory and excitatory activity may lead to altered signaling to second-order neurons in the intermediate zone which, through an excitatory polysynaptic chain, excite motor neurons in the ventral horn to initiate protective movements or abnormal proprioceptive behaviors. Such abnormal sensory processing is suggested at least for thermal stimuli, as Gbx1 mutant males displayed increased latency suggesting reduced pain in the hot plate test (thermosensory functions). Finally, considering that locomotor deficits become apparent at P16-17, we cannot exclude that abnormal gait may result from postnatal developmental or neurodegenerative events, which would need to be investigated.

Despite the clear behavioral phenotype and reduced pool of GABAergic neurons in the dorsal horn, we did not observe any change in the expression of homeodomain factors involved in dorsal spinal cord patterning, or markers for primary sensory afferents, indicating that the development of the dorsal horn is not profoundly affected in Gbx1−/− mutants. An explanation for these results—and for the overall mild phenotype of the mutants—is that Gbx1 and Gbx2 are coexpressed in dorsal spinal cord cells at early stages of embryogenesis: hence the presence of Gbx2 (and its subtle upregulation observed at E12.5–E14.5 in mutants) might compensate for Gbx1 loss of function with respect to early regulatory events. Generation of Gbx1;Gbx2 double mutants will be required to assess possible redundant functions, and the availability of a Gbx2 floxed (conditional) allele does allow strategies for a spinal cord-specific inactivation, which would alleviate the lethality of the Gbx2 null mutants (Wassarman et al., 1997).

Despite the importance of dorsal spinal cord in normal sensory processing, our knowledge concerning the establishment of neuronal circuits remains limited (Graham, Brichta & Callister, 2007; Todd, 2010). In this regard, our work contributes to understanding how transcription factors cooperate for regulating cell specification and eventual distribution of neuronal subtypes in the developing spinal cord, providing clues for further dissecting functional circuitry of the dorsal spinal cord.

Supplemental Information

Figure S1 Expression analysis of Gbx2 in the developing spinal cord of Gbx1 mutants

Sections through the spinal cord of wild-type (A, C, E, G) and Gbx1−/− (B, D, F, H) mice are shown. All sections are at the lumbar level. In situ hybridizations for Gbx2 were performed at different developmental stages: E12.5 (A, B; n = 2), E14.5 (C, D; n = 2), E16.5 (E, F; n = 3), and E18.5 (G, H; n = 3). Scale bars: 100 µm.

Click here for additional data file.

Figure S2 Analysis of rhombomeric markers in Gbx1−/− embryos

Whole-mount in situ hybridizations of E9.5 embryos with 2 markers of prospective rhombomeres: Hoxb1, which labels rhombomere 4 (A, B; n = 3), and Hoxa2, which marks rhombomeres 2 to 6 and associated neural crest (C, D; n = 3). Scale bars: 50 µm.

Click here for additional data file.

Figure S3 In situ hybridization analysis of Gad67-expressing cells in the prenatal hindbrain

Sections are shown at various levels of the brain stem (A–F) and cerebellum (G, H) of wild-type (A, C, E, G; n = 3) and Gbx1−/− (B, D, F, H; n = 3) mice at E18.5. Scale bars: 100 µm.

Click here for additional data file.

Figure S4 Analysis of developing spinal cord motor neurons in Gbx1 mutants

Expression of Islet1 in the lumbar spinal cord of wild-type (A, C) and Gbx1−/− (B, D) mice at E14.5 (A, B; n = 3) and E16.5 (C, D; n = 3). (E) Countings revealed that the numbers of Islet1 + cells in the ventral horn are not significantly diminished in Gbx1−/− mice (at E14.5: 76 ± 7.33 Islet1+ cells in WT; 75.22 ± 3.13 in Gbx1−/− mice; Genotype F(1, 4) = 0.27, NS, Sections F(2, 8) = 0.18, NS, Genotype*Sections F(2, 8) = 0.27, NS; at E16.5: 22.38 ± 5.96 Islet1 + cells in WT; 25.33 ± 6.70 in Gbx1-/- mice; Genotype F(1, 4) = 3.03, NS, Sections F(2, 8) = 4.73, p < 0.05, Genotype∗ Sections F(2, 8) = 5.46, p < 0.05). Scale bars: 100 µm.

Click here for additional data file.

Video S1 Supplemental Movie

Click here for additional data file.

We thank B Schuhbaur for excellent technical assistance. We are grateful to Dr. K Niederreither for a critical reading of the manuscript, and to Drs. V Brault, M Paschaki and D Dembélé for help with statistical analysis. We thank Drs. C Birchmeier, S Britsch, F Chen, K Jagla, P Bouillet, B Giros, P Gruss, M Tessier-Lavigne, R Krumlauf, F Rijli and AJ Tobin for the gift of reagents.

Additional Information and Declarations

Competing Interests

Author Contributions

Animal Ethics

The authors declare no competing financial interests. PD is an Academic Editor for PeerJ.

Hamid Meziane performed the experiments, analyzed the data, wrote the paper.

Valérie Fraulob, Fabrice Riet and Michaela Geffarth performed the experiments.

Wojciech Krezel conceived and designed the experiments, analyzed the data, wrote the paper.

Mohammed Selloum and Yann Hérault contributed reagents/materials/analysis tools.

Dario Acampora performed the experiments, analyzed the data.

Antonio Simeone and Michael Brand conceived and designed the experiments, contributed reagents/materials/analysis tools.

Pascal Dollé conceived and designed the experiments, wrote the paper.

Muriel Rhinn conceived and designed the experiments, performed the experiments, analyzed the data, wrote the paper.

The following information was supplied relating to ethical approvals (i.e., approving body and any reference numbers).

Animal experimentation protocols were reviewed and approved by the Direction Départementale des Services Vétérinaires (agreement #67-172 to HM, 67-189 to PD, and institutional agreement #D67-218-5 for animal housing) and conformed to the NIH and European Union guidelines, provisions of the Guide for the Care and Use of Laboratory Animals, and the Animal Welfare Act.

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
