# Peer review of "The homeodomain factor Gbx1 is required for locomotion and cell specification in the dorsal spinal cord"

_PeerJ, doi:10.7717/peerj.142_

## Round 0.1 · original submission · Major Revisions

I have now received 2 reviewer’s reports for your manuscript and reviewed it myself. Overall, I find the work to be valuable, but there are some issues that I think are important to be addressed before the manuscript is accepted for publication. While some comments suggest that the work would benefit from additional experiments, I understand this is not always possible. However, I think the issues raised are valid and should be at least addressed in the text of the manuscript.

General Interpretation of the data:

Gbx1 -/- phenotype: One of the reviewers asks whether there might be a compensatory expression of Gbx2 as a result of loss of function of Gbx1. I think that it would be useful to address this in the discussion.

Anatomical/histological analysis: It seems the data for ICChem and ISH is based on a single time point which is also different from the age at which motor function was assessed. Both reviewers highlighted the limits of how these data can be extrapolated to relate it to the behavioural results. One reviewer asks what the expression patterns observed with ICChem and ISH might be in the spinal cord in more mature rats to warrant the link made between GABAergic phenotype and behavioural deficits. I would agree that showing that the differences observed in embryos persist beyond the age study would be useful. One reviewer and I also wonder about the use of ISH GAD-67 used as a proxy for GABA-ergic phenotype (suggesting the use of a second marker such as Pax2). I also wonder what the expression patterns of GAD-65 might be. Is it possible that those neurons that are not expressing GAD-67 might still express a GABAergic phenotype? Or is it possible that the differences observed might reflect a difference in the time course of expression of the GABAergic phenotype? One of the reviewers also queries whether the onset of locomotor deficits with respect to the transition of GABA between inhibitory to excitatory. Could this be added to the general discussion?

One reviewer also queries whether the lack of obvious abnormalities in the hindbrain at E9.5 is sufficient evidence to rule this possibility out. Is the GABA phenotype, for example, normal in hindbrain at later stages, or are there deficits similar to those described in the SC? I also wonder whether it might be useful to show the hindbrain data on which these conclusions are based. It seems that this is important data that is substantial to the interpretation of the behavioural data (as highlighted in pp. 12-13).

Similarly, on P 14 the statement is made that “Altogether, these data suggest that there are not defects in patterning…” My impression is that what is provided is a very broad morphological analysis on a single time point. I am weary as to the degree of confidence with which that statement can be made, eg., do these data show with enough detail layer-specific projections, and is a single time point sufficient to rule out defects in patterning?

Pain sensitivity: One reviewer questions the interpretation of the possible role of allodynia. I also wondered why the argument is made that the Gbx1 -/- might have reduced thermal pain sensitivity – an alternative is that they have issued processing reflexes involved in thermal pain which may not necessarily be accompanied by a reduced sensitivity per se.
In the results, the statement “showed a tendency for higher latencies than WT, but the difference between genotypes was not significant” should be corrected. If there is no statistical difference, then I don’t think the implication that the latencies may be higher is not justified. Similarly on Page 12 “at least Gbx1 -/- females [..] tended to have higher number of slips” these differences were not shown to be significant and “Gbx1-/- mutants displayed impaired rotarod behavior […]” but table 1 and p10 this is described as NS.

General Methodological queries and replicability.

Both reviewers identify areas where further methodological information would be useful for the interpretation of data, and further descriptions are needed to ensure replicability of these studies.

Please provide the N for all experiments- for example, tissue collection and sample preparation does not say how many pregnant females were used and how many embryos from each were harvested. This is important when considering the actual size of the sample analysed for histology and to infer possible litter effects in the results.

It was also not clear to me in the behavioural phenotyping experiments whether the cohorts described mean that each cohort was put through a given paradigm, or whether there were several cohorts each tested through the set of behavioural paradigms.

Please provide a description of how the blastocysts were obtained (Page 6, second paragraph).

Please indicate whether behavioural experiments were performed during the day or night cycle of the rats.

Cell counting: One of the reviewers queries about how GAD-67 + cells were counted, and whether the images shown are representative of the data. Cell counts are described as being made in 9 sections for each animal and 3 animals of each genotype. I am assuming the data in the plot in Figure 8 represents an average of the 27 sections for each genotype, but this could lead to pseudoreplication. The n for each genotype should be 3, each represented by the average of the 9 sections (which would probably preclude from using the parametric t-test). It is also not reported whether the tissue used for this study was from males and females, and given the male-female differences seen in some of the behavioural tests and the sensory conduction velocity this is an important factor to take into account.

Assessment of motor deficits: One of the reviewers queries about what is meant by “duck type gait” and “lack of fluidity” in movement. I wonder whether providing a movie (if available) might aid in the description. The reviewer also queries it would be useful to look at time s. pent in extension and flexion.

Electromyography: The description of the electrode placed at the “base of the tail” is not too clear. I am assuming since it is described as an EMG it would be a muscle, but I fail to see how recording from a muscle would provide a measure of the sensitive (do you mean sensory?) nerve conduction velocity. I am also not sure how these latencies were actually measured. How much precision was there in the placement of the stimulating electrode at 20 mm away from the recording electrode? It is also not stated how the data was analysed – how many animals were used, how many recordings per animal, etc. Would be worthwhile to show an image of these recordings, especially given the differences between males and females?

General corrections:

Reference to Table 1 at the bottom of Page 11 should be Table 2? There is also a discrepancy with the text saying females had increased conduction velocity, but in the table it marks the males as being significantly different. Similarly re: latency (top of p 12) the text says there is N.S. but the table suggests there is a significant difference in females. Results for rotarod need to be checked for accuracy. The statement “males and females tended to have reduced locomotor activity over the testing period (p=0.09). Is this p value correct? Figure 7 also needs correcting.

General suggestions

I am surprised that the sexual dimorphism in the Gbx1-/- phenotype is not addressed. Would it be useful to speculate on its significance?
In the introduction (first paragraph) tactile projections are described as projecting to laminae III, IV and V and temperature and pain as projecting to lamina I, II and III, whereas in page 14 the former are said to project mainly to laminae III, IV and V and the latter to lamina I/II. Perhaps it should be made clear in the introduction the degree of projections to lamina III from pain and temperature receptors.

I am not sure I follow the argument on P12 “Gbx1-/- females had increased sensory nerve conduction velocity [..] supporting the altered sensory deficits observed in the behavioural tests” nor in P14 paragraph 2, I suggest revising the wording.

P14 par 1 Can you provide references for the statements on the second half of the paragraph?

Please add the names of people providing supplies (for immuno and in situ) to the acknowledgements and obtain permission for that. Please declare if these supplies would be available for replication studies.
Animal Ethics Statement is presented under the generation of genotyping of chimeric and mutant mice – I would assume that ethical approval has been sought for the entirety of the project. If so, it might be made more explicit by placing the animal ethics statement at the beginning (or at the end) of the M+M.

.

·

Basic reporting

- In figure 1 it would be interesting to look for the expression of Gbx2 to see if there is a compensatory expression of Gbx2 due to the loss-of- function of Gbx1 in mutant mice, especially in developmental ages from E12.5 onwards.
- No obvious developmental abnormalities were found at the hindbrain in E9.5 Gbx1-ko mice. The authors should look for alterations at later developmental ages, using terminal differentiation markers (as it was done at the spinal cord level) to completely exclude the involvement of hindbrain in the observed mutant phenotype.
- The authors discuss that allodynia cannot be discarded as a cause for the altered gait phenotype. In the hot plate analysis, Gbx1 ko mice (particularly the males) displayed a reduced response in hot plate test indicating that mutant mice feel less pain. Therefore, allodynia does not seem a possible scenario to reconcile with the observed hot plate performance.
- In the characterization of the spinal cord dorsal horn of ko mice, there are several weaknesses that compromise one of the main conclusions of the manuscript, which concerns the cell specification defect in the dorsal spinal cord (see at "Experimental Design").
Minor corrections:
- Figure 7 C,D and E,F does not correspond to text description.
- In the text the results of increased sensory nerve conduction velocity are stated in Table2 and not in Table 1.
- In Table 1 – rotarod, results for Gbx1-KO females does not seem correct.
- On page 4 it is mentioned that Drg11 is expressed in late born derived cells from dI5 neurons but the reference mentioned is not fully correct. Authors should consider adding the studies by Rebelo et al (2010 Dev Dyn).

Experimental design

1- It is not clear how counting of Gad67-positive cells was performed (a full description is need in the Materials and Methods section) and, in figure 8, Gad67 in situ staining pictures do not reflect the cell counting shown in the graph. As the difference is not so marked, the authors should use other GABAergic markers, such as Pax2, to corroborate the result of a reduction of GABAergic neuronal population.
2- If these Gad67-positive neurons are missing, it is important to address whether these neurons died, change to a glutamatergic phenotype or possibly there is simply a Gad67 expression down regulation. Therefore, neuronal death markers and glutamatergic markers should be looked up. Indeed, it may be relevant to double check a possible difference in the Drg11 expression (Figure 7 EF) since at sight there seems to be a difference in expression between wild type and Gbx1-knockout embryos at E16.5.
3- It would be also important to address post natal expression abnormalities at the spinal cord level atfer the first two or three weeks after birth.

Validity of the findings

- The authors conclude that Gbx1 has a role in the specification of a subset of GABAergic dorsal horn interneurons involved in the control of hindlimb movement, however it’s not clear how this assumption could be made. It is well established that limb movement is controlled by neuronal circuits located at ventral spinal cord and that somatosensory information is integrated and conveyed to the brain by the dorsal spinal cord. Therefore, it’s no clear how authors could do such a conclusion.

Additional comments

In order to find a physiological function for Gbx1, the authors generated and characterized the phenotype of the Gbx1-knockout (ko) mice at two levels: molecularly during development and behaviorally. After a battery of behavioral tests the most striking phenotype was the altered gait during forward locomotion affecting the hindlimbs. The authors then proceeded trying to correlate this observation with a molecular defect at the neuroaxis.
This paper requires, however, significant revision.

Reviewer 2 ·

Basic reporting

It seems odd to end the introduction with a reference based on the content of the sentence?

Experimental design

No comments

Validity of the findings

pg. 10- the description of a "duck type" gait needs explanation. It was not immediately apparent what was meant by this. Describing the specific sequence of hyper-flexion and hyper-extension behaviors would be more useful to the reader.

pg. 10- For the following statement "However, many of the Gbx1-/- mutants showed significantly abnormal gait (χ2 ≥5.20, p<0.05). Indeed, 43% of Gbx1 mutant males and 63% of Gbx1 mutant females displayed lack of fluidity in movement" clarify the test that was used to demonstrate gait was abnormal and motion was not fluid.

Additional comments

1) It would be interesting to determine whether behavioral/locomotor deficits coincide with change of GABA from excitatory to inhibitory which (I think) happens around P17 in mouse- well after the onset of weight bearing locomotion. If the locomotor deficits began around this time it would really strengthen the paper by directly linking them to GABA.

2) The conclusion really needs a discussion of the experiments required to determine the potential source of the locomotor deficit (central vs proprioceptive deficit). Some discussion of the locomotor CPG (role of other genetically defined populations and how they might be connected to GABAergic neurons) as well as the role of proprioceptive input in generating locomotor outputs needs be included in this section.

3) It would be useful to quantitate the amount of time spent in flexion (paw raised) vs time spent in extension (paw on ground) and compare to wt. In "regular" mammalian locomotion as speed of locomotion increases the proportion of time spent in the flexor phase (swing) increases. Is this the case in the Gbx1-/-?

---

## Round 0.2 · Major Revisions

I have now gone through the revised manuscript and my comments are listed below. My overall impression is that some of the issues that were raised during the first round of review still need to be further addressed. My comments below reflect my views on the current version as well as how well I consider the issues raised by the first round of review were addressed.


Results reporting:

1. In response to Reviewer’s #1 request to address compensatory expression of GBx2, Fig S1 appears to satisfy that request. However, the data is interpreted as only showing a “possible subtle increase at E14.5”, but in the images provided there also seems (to me) to be an increase in expression at E12.5, and some staining pattern differences at other ages. Were these data quantified or was this a qualitative analysis. If the latter, what were the criteria used to assess differences between and within groups? Can the authors please comment on the possible implications of these differences?

2. Report on motor performance: Paragraph 3 states that in the beam walking test the latency to cross the beam was significantly increased, but the figure shows this to be true only in females. I suggest this be qualified in the text. Similarly “the number of slips was slightly increased” is misleading, since it is not significantly different. “Males were less affected than females” suggests that males are affected, which is not what the data shows. I understand your suspicions regarding the power of some of the analysis, the analysis provided shows no significant differences, and it might be worthwhile to design a repeat of the experiment with more power to determine whether these tendencies are significant or not. Similarly in paragraph 5 of discussion: “tended to have higher number of slips” need to qualify this was NS.

3. Report on motor performance: “males and females tended to have reduced locomotor activity […] p=0.09 . Isn’t this NS?

4. Report on motor performance: “reduced pain sensitivity” Not sure the sensitivity to pain, but rather the response to pain is what is measured. If the argument is made that it is the reduction of GABA cells in the DSC that are responsible for the motor deficits, I am not sure how this relates to reduced sensitivity at the primary afferent level – I still fail to see why the conclusion is that there are proprioceptive defects and/or altered sensory abilities. Are there no alternative explanations? Though this might be simply a semantic issue. The way I read it is that the argument is made that the deficits are in the sensory periphery/primary afferents.

5. Hindbrain analysis: In response to Reviewer #1 an analysis at E18.5 was performed in the hindbrain using GAD67 and reported as no difference between wild type and mutant – Figure S3 G,H appears to show a reduced expression of GAD67 in cerebellum., and perhaps some increase in the mutants in dorsal hindbrain (Panels E/F). Of course, I am not able to look at the entire section set. Was this quantified, or is this a qualitative assessment? If the latter, were any criteria used?

6. Development of spinal cord dorsal horn:The text says that Nissl staining of E18.5 mice … (Fig 6) but the Figure legend says it is E 16.5.

7. For figure 4 the text states that there are “no defects in patterning of sensory afferent fiber projections” but this conclusion is based mainly on the staining of a subset of sensory afferents (that are labeled with calbindin) and on the apparent lack of differences in expression of Drg11. I think the resolution of the images in panels A and B is too low for the reader to assess any possible differences and/or ectopic projections. Also, the markers used cannot rule out other types of patterning differences from primary afferents that do not label with calbindin or possible patterning differences at ages other than the one provided. The statement that follows regarding that males show longer withdrawal to the hotplate cannot be easily correlated with the images shown since it is not known whether the tissue presented corresponds to males or female animals, and this limitation should be made explicit. It would be useful to refer to Buckley et al.’s results that indicate that indeed there are abnormal proprioceptive projections.

8. In panels EF of Figure 4 the text reports no differences in the patterns of staining for peripherin. There appears to be a down regulation of peripherin at least in the most medial ventral spinal cord, a finding that is consistent with those of Buckley (2013). Were you able to find ectopic projecting axons such as those described by Buckely et al?

9. The statement that “no differences in the number of Islet1+ cells were found in the ventral spinal cord - was unable to find this quantification – or was this a qualitative assessment? If so, can you please describe how this was done.

10. I am also surprised that the differences in ages between the Buckley study and the present study are not considered as a source of discrepancy.

11. Regarding Reviewer#1 request to look at the fate of neurons that do not become GAD67 positive – Is there any reason why the apoptosis analysis is not shown or details of how this was analysed provided?

12. In response to Reviewer #1’s request to address post natal expression abnormalities at the spinal cord level, the argument is made that these will be part of a separate study. While I understand why the choice of separating these two studies may be appropriate, it does not address the reviewer’s concerns regarding how appropriate it is to draw causal conclusions on mature motor performance based on embryonic GABAergic phenotype. The authors might want to consider discussing this limitation in more detail in their discussion.

13. In response to reviewer #2’s query about the onset of the motor deficits, you provide a statement that the deficit first appears at P16-17. Could you please provide details on how this was analysed.

14. In response to the editor’s summary you state that you have repeated analyses at several developmental stages. I am unable to find those data, other than for the additional material provided in the supplementary figures (for the additional Gbx2 expression, the analysis of hindbrain and the expression of Islet in the SC). However, the main ISH and ICChem data on which the main conclusions are based seem to only be described at a single time point.

Methods reporting:

15. Can you please provide a reference to the methods used for obtaining the blastocysts, etc. – I understand this might be standard, but I would imagine the source would be needed for any attempts of replication.

16. In response to Reveiwer#2’s request to clarify how abnormal gait was assessed, I am unable to find a description in the material and methods for other groups to be able to replicate your work. Could you please provide the details including what criteria were used for the assessment of what was “normal” or not. I am assuming observers were familiar with normal gait patterns in normal mice prior to the observational classification? Did both observers always agree on the classification?

17. In response to Reviewer #1 regarding details on the cell counting method: I still don’t think that enough detail has been provided for an independent replication of the data. The figure legend says the areas shown in panels C and D were the ones used for counting – but there are areas where there is no tissue included in those areas – Was there a criterion as to where the counting box was placed, that is, over what anatomical area were the cells counted? Also could you please provide information as to the objective used (eg NA) to get a sense of resolution, and whether the cells counted under the microscope or from printed photographs? Like the reviewer, I still fail to see in the images the reduction that is reported in the graphs.

18. It also seems to me that if one is counting cells over a defined area (which according to your figure excludes some tissue where GAD cells are present too), what is being provided is an estimate of cell density rather than actual cell number .

19. I am also confused in the graph what the % of cells refers to, and hence what the reported percentage reduction refers to. Can this be clarified further please? Also, if 3 sections of 3 animals were used for counting for each, the n for each group would be 3, not 9 – I am not convinced a t test is appropriate, and wouldn’t think it is appropriate to plot the data as means and standard deviations. These issues also apply to the quantification of Pax2 and VGLUT.

20. The mouse anti-islet1 antibody is reported as being used at 1:100 – please indicate whether this was the supernatant or the concentrate form. Please provide the concentration used for the secondary antibodies. Can you please provide the constitution of the buffers in which antibodies were diluted and washes made. Could you please also provide (or specify) your experimental controls results.

21. Rotarod test: I am not sure what the “latency” in the test is. Am I right to assume this is the time it takes for them to fall? Please specify.

22. Electrophysiological measures: I am still not sure how the stimulating electrode was placed on the sciatic nerve, or how the distance between the stimulating electrode was measured (and with what accuracy). I am also no sure how these latencies were actually measured or how the data were analysed. I am also unsure how many recordings per animal were done, etc. Can you please also provide the dosage of anaesthesia.

There are also a few typographical/grammatical errors in the manuscript.

---

## Round 0.3 · Minor Revisions

Please address the following remaining issues, as well as other issues as per our offline correspondence. Please note that the reviewer's comments are incorporated into the Editor's comments, so there is no need to respond to those separately.

1. Can the authors please comment in the manuscript on the possible implications of the subtle increase of Gbx2 expression at E12.5-E14.5

2. Reviewer #1: Quantifications of number of stained cells for each molecular marker displayed in fig 8 should be accompanied by an appropriate statistical analysis to assess the significance of the differences observed.

If the authors can argue why this statistical analysis cannot be performed, these limitations, and the limitations on the interpretation of the data need to be made explicit.

3. To address the query about the possible fate of GABAergic neurons the authors complemented their GABAergic phenotype studies with examination of a glutamatergic marker at E18.5 and a cell death marker (at E12.5, 14.5, 16.5, 18.5). I think that it would be valuable to include these figures in the main article rather than in the supplementary material.

4. Could the authors also please discuss how the ages at which the TUNEL was examined correspond to expected apoptosis through naturally occurring cell death in the mouse lumbosacral spinal cord and how this may (or may not) have an impact on your data results and interpretation.

5. To address the concerns about how valid it is to compare the embryological data to the adult phenotype, the authors have added to the histological examination at E18.5 the observation that the onset of defects coincides with the switch of GABA from excitatory to inhibitory. These observations are described as being obtained by checking pups visually around weaning. Can you please provide the specific ages and frequency with which this monitoring was done, the number of animals and if possible the male to female ratio. Also, please double check reference to Ben-Ari et al. “which coincides with the time point at which the GABA pathway switches from excitatory to inhibitory (Ben-Ari et al. 2007).” I was unable (albeit through a quick superficial scan) to find in that paper mention of a switch from excitatory to inhibitory in mice spinal cord at that age.

6. Reviewer #1: Moreover, the present data differ from the Buckley’s data in that the expression of peripherin and Islet1 was not changed in the Gbx1 -/- model. This discrepancy was attributed to the use of different time points. Although I don’t think this may be the reason, if the authors insist in this explanation they should evaluate the expression of those markers at the same time points. In addition, it would be very interesting to analyse their expression at postnatal stages, when the locomotor defect is detected, as requested on our previous revision
6.1. Regarding the discrepancies between this and Buckley’s results on the peripherin staining, it might be useful if the authors used an image of peripherin at E16.5, since they already have that material and the age is closer to that of Buckley et al. (E14.5 and E15.5) to allow for a better age-matched comparison. I think this would satisfy the reviewer’s request.
6.2. Regarding the discrepancies in Islet1+ results between the current work and that of Buckley et al., please amend the statement: ” In contrast to our observations, those mutants show disorganized peripherin expression, together with a decrease of Islet1-expressing cells in the ventral horn of the lumbar spinal cord (Buckley et al. 2013).” See Figures 9 (showing change at E14.5-15.5) and Fig 5 (no change at E11.5). Since Buckley et al. show a decrease of ISL1 cells at E14.5 and E15.5, it might be useful to include images and the quantification for E16.5 in your material for a better age-matched comparison. If you do not have that quantification, please amend the sentence “No differences in the number of Islet1+ cells within the lumbar ventral spinal cord were found at E14.5, E16.5 or E18.5” by removing the E16.5.

7. For figure 7 the text states that there are “no defects in patterning of sensory afferent fiber projections” but this conclusion is based mainly on the staining of a subset of sensory afferents (those labeled with calbindin) and on the apparent lack of differences in expression of Drg11. I think the resolution of the images in panels A and B is too low for the reader to assess any possible differences and/or ectopic projections, and cannot differentiate between afferent inputs and neuropil stained originating from what appear as calbindin+ neurons in the SC. Also, the markers used cannot rule out other types of patterning differences from primary afferents that do not label with calbindin or possible patterning differences at ages other than the one provided. Please address these limitations.

8. Reviewer #1: On page 9 the authors say “Altogether the behavioural data revealed that the Gbx1 -/- animals ….because mutant mice also display a significantly reduced response time in the hot plate (thermosensory) test”. This paragraph should be revised correcting the hot plate test observation (underlined) and should be better explained how animals with less pain sensitivity in the hot plate could suffer from pain on movement.

9. Reviewer #1: In the conclusion section, in the end of first paragraph the sentence “… as mutant mice also display a significant reduced response time in the hot plate (thermosensory) test.” Is in contradiction with the affirmation in the third paragraph “… as Gbx1 -/- mice displayed significantly increased response time in the hot plate test.”

10. The sentence “We cannot exclude that the proprioceptive defects may be secondary to a defect in sensory pathways (for example, caused by pain on movement), because mutant mice also display a significantly reduced response time in the hot plate (thermosensory) test.” Should be corrected to reflect that only males showed a significant difference and this was an increase (not decrease) in the response latency.

11. In resonse to the request for details on the methods by which abnormal gait was assessed, please indicate whether observers were familiar with normal gait patterns in normal mice prior to the observational classification

12. [R#1: In the Buckley study as in the study under revision, it is not discussed how a defect in dorsal spinal neurons could interfere with the proprioceptive afferent system. The authors should address this point in the discussion, as requested in our first revision of the paper. ]
12.1. The authors might also want to make explicit that changes in the GABA phenotype were not analysed outside of the dorsal horn, and that contribution of inhibitory circuits elsewhere in the spinal cord cannot be excluded.
12.2. In the statement: ““We cannot exclude that the proprioceptive defects may be secondary to a defect in sensory pathways (for example, caused by pain on movement), because mutant mice also display a significantly reduced response time in the hot plate (thermosensory) test.” Please amend the sentence to reflect that only males displayed a significantly reduced response time.
12.3. In the statement: “Altogether, the behavioral data revealed that the Gbx1-/- animals have apparent proprioceptive defects and/or altered sensory abilities.” I am still uncertain as to how after failing to show significance on sensory features measured (other than conduction velocity in females and hotplate in males) this statement can be justified.

13. The statement “This is supported by data from the beam walking test showing that at least Gbx1-/- females required longer time to cross the beam distance and tended to have higher number of slips (not statistically significant)” if it is NS it is not supported by the data – the trend you see might indicate that a higher power test might uncover this, but it is speculation.

14. The statement: “Electrophysiological measurements showed that Gbx1-/- females had increased sensory nerve conduction velocity, measured in the caudal nerve, supporting the altered sensory functions observed in the behavioral tests.” Not sure what sensory functions are being referred to here. Responses in the hot plate, startle, beam slips and sensorimotor all failed to show an effect on phenotype.

15. The statement “We found that both Gbx1-/- males and females show reduced locomotor activity in different situations.”, but legend to figure 4 says “The distance traveled over the 20 min period of test reflects locomotor activity” and shows no difference, and on page 11 this is described as “. When considering each gender separately, both Gbx1-/- males and females tended to have reduced locomotor activity over the testing period (although not statistically significant, p=0.09) (Fig. 4).”

General corrections:

16. R#1: In results section when describing Drg11 expression they should mention (Fig 7C,D and data not shown)
16.1. I found similar instances throughout the manuscript. Please differentiate between the data that was used for the analysis but that is not shown and that which is shown: Lbx1 analysis reported at E12.5, 14.5, 16.5, 18.5 but only shown at E16.5. Lmx1b analysis reported at E12.5, 14.5, 16.5, 18.5 but shown at E16.5; etc. In the text: “or at the level of fibers that enter into the spinal gray matter, at E16.5 or E18.5 (arrows in Fig. 7E,F).” not clear which age is shown in the figure.

17. P16: The sentence “Importantly, Gad67 expression was reduced in the dorsal spinal cord of Gbx1 mutant mice (Fig. 8A-D), i.e. there was a 16% decrease in the number of Gad67-expressing cells in comparison to WT mice (Fig. 8I).” should be corrected – there was a 16% decrease in the proportion of GAD67 expressing cells. The same applies to sentence “This finding was strengthened by analysis of Pax2, another gene expressed in GABAergic cells in the spinal cord (Cheng et al. 2004), with cell countings corroborating the decrease in the number of GABAergic cells” Similarly in the legend to Figure 8 the statement “Countings (percentages of labelled vs. total cells) revealed that the numbers of Gad67+ cells are diminished by 16% in Gbx1-/- mice” should be corrected to “revealed that the proportion of GAD67+ cells”. The same applies to “number of Pax2 cells” and “numbers of Slc17a6+ cells”

18. There is a discrepancy in the reporting of the Netrin-1 material – the text says the image shown is E18.5 whereas the figure legend suggests it is E16.5

19. On Figure 7, Please specify the age at which the Drg11 and Peripherin staining shown was performed.

20. P11: “checked the expression of Gbx2 from E12.5 onwards” specify ages
21. P11: “unnevenless” check spelling

References
22. “Waters, Wilson & Lewandoski 2004”. Please double check – PubMed cites as Gene Expr Patterns. 2003 Jun;3(3):313-7.
23. Reference to Nagy et al. 2003 does not seem to be in the reference list

24. In order to adhere to PeerJ’s editorial policy: “Methods should be described with enough information to be reproducible by another investigator” and that “the data should be robust, statistically sound, and controlled” I will ask the authors to make the following amendments:

1. Further details on how cell counting was performed need to be provided – eg what does using ImageJ mean? Was a threshold applied for counting, or just used to track cells? Were the sections counterstained, and if not, how were the unlabelled cells visualised. Unlike the case of the material shown in S4, it is difficult to see from the images provided how the authors were able to count labelled and unlabelled cells.
2. Please make sure that all measurements in the tables and graph axes are accompanied by the appropriate measuring units
3. Please try to track down which of the two forms of the DSHB Islet1 antibody was used for these studies
4. Please go through the M+M to make sure all buffers and solutions are specified as well as possible (e.g., P6- L2: PBS what molarity?; P6- L4: PFA in what buffer?; P6- L4: 20% sucrose in what buffer?; P7- L2: dehydrated how?; 3% hydrogen peroxide in what buffer, etc)
5. Please provide surgical, electrodes etc and other details used for the electrophysiological studies.
6. In all figures (or figure legends), where appropriate, please report the N, and whether the data represents mean/median, SD/SEM etc.

Reviewer 1 ·

Basic reporting

We are pleased to verify that the authors have addressed several of our comments, which contributed to increase the quality of the manuscript.
However, very recently another study on the characterization of a Gbx1 -/- allele was published (Buckley et al, 2013) demonstrating a a locomotive defect affecting the hindlimb gait and stating that this phenotype results from abnormal formation of the proprioceptive sensorimotor circuit. In the Burckley’s study, as in the study under revision, it is not discussed how a defect in dorsal spinal neurons could interfere with the proprioceptive afferent system. The authors should address this point in the discussion, as requested in our first revision of the paper.
Moreover, the present data differ from the Burckley’s data in that the expression of peripherin and Islet1 was not changed in the Gbx1 -/- model. This discrepancy was attributed to the use of different time points. Although I don’t think this may be the reason, if the authors insist in this explanation they should evaluate the expression of those markers at the same time points. In addition, it would be very interesting to analyze their expression at postnatal stages, when the locomotor defect is detected, as requested in our previous revision.
Minor comments:
- Quantifications of number of stained cells for each molecular marker displayed in fig 8 should be accompanied by an appropriate statistical analysis to assess the significance of the differences observed.
- On page 9 the authors say “Altogether, the behavioral data revealed that the Gbx1-/- animals … because mutant mice also display a significantly reduced response time in the hot plate (thermosensory) test.” This paragraph should be revised correcting the hot plate test observation (underlined) and should be better explained how animals with less pain sensitivity in the hot plate test could suffer from pain on movement.
- In results section when describing Drg11 expression they should mention (Fig. 7C,D and data not shown)
- In the conclusion section, in the end of first paragraph the sentence “…as mutant mice also display a significant reduced response time in the hot plate (thermosensory) test.” Is in contradiction with the affirmation in the third paragraph “…as Gbx1-/- mice displayed significantly increased response time in the hot plate test…”.

Experimental design

Nothing to add

Validity of the findings

Nothing to add

Additional comments

This is an interesting paper that would benefit with the ameliorations suggested

---

## Round 0.4 · Minor Revisions

Thank you very much for the resubmission, and also for your effort addressing the issues raised. I want to once again emphasise that I consider this to be a valuable set of experiments that deserves to be published, and I hope you will be able to address some remaining issues that prevent me from accepting the manuscript at the time.

One important issue still remains regarding the analysis of the GABA cells, which is of great importance since this seems to be at the centre of the interpretation of the behavioural phenotype.

1. I was surprised that Dr Dembele suggested a paired t-test since I couldn’t understand what criterion/justification was used to pair the animals from the two genotypes (and I couldn’t find where one was provided). I have consulted with a colleague of mine in the statistics department (Dr Russell) who suggested that since you have 2 genotypes and 3 observations in each of 3 animals per genotype, the data would be best analysed with a 2 way mixed model anova with the first variable as treatment effect (fixed effect ie, genotype_ and a second variable as random effect of repeat observations on the same individual. He provided me these resources
http://www.statisticshell.com/docs/mixed.pdf and http://www.math.montana.edu/~cherry/st412/pdf_files/CourseNotes16.pdf
Please note that the same criticisms apply to the data on Figure S4.
2. I am also surprised that the authors were unable to find any references to the switch of GABA from depolarising to hyperpolarising to speculate at what age this might be happening in the dorsal horn at the levels of the spinal cord where the GABA phenotype is described. My search returned a few articles that the authors may want to consider: Stein, V., Hermans-Borgmeyer, I., Jentsch, T. J., & Hübner, C. A. (2004) J Comp Neurol 468(1), 57–64 sets the shift at around P7-P10 at least for hippocampal and spinal cord motoneurons in mouse; Cordero-Erausquin et al (2005) J Neurosci 25(42):9613-8623 show similar time course for the switch of GABA in neurons of LI in the dorsal horn (albeit in rats) while Sibilla, S., & Ballerini, L. (2009). Progress in Neurobiology, 89(1), 46–60 report these changes as happening within the first two weeks of age. This would suggest (if I have interpreted the articles correctly) that the switch may be over by the time that the phenotype becomes expressed (P16-17). The authors may also want to consider when incorporating this to their discussion that a depolarising effect of GABA does not necessarily means it is excitatory (some depolarising GABA has been shown to be inhibitory, eg. Viemari, J.-C et al (2011). Importance of chloride homeostasis in the operation of rhythmic motor networks. In Progress in Brain Research (Vol. 188, pp. 3–14). ).
3. The statement : “As expected, natural cell death occurs in the developing spinal ganglia (Fig. 9A, arrow; Paschaki et al. 2012) but not in the spinal cord” and what follows:
The reference does not support the statement, since it does not constitute a study of normal cell death in mice, and, within that same reference it appears to me that apopototic labeling in the ventral spinal cord at 12.5 can be seen (Figure 4I). There are a number of studies looking at normal cell death in mouse spinal cord, and my interpretation of some of those is that one would expect to see cell death in the spinal cord as well as the DRG at least in your E12.5 and E 14.5 material. (eg, White et al Journal of Neuroscience, February 15, 1998, 18 (4):1428); Yamamoto, Y., & Henderson, C. E. (1999). Developmental biology, 214(1), 60–71). It would be useful to discuss why it might be the case that no TUNEL labeling is visible in your material at those ages, which is crucial to validate the results.
4. I will also ask that those data where the statistical analysis failed to show a statistical significance are clearly described as such. Eg the sentence : “Such abnormal sensory processing is suggested at least for thermal stimuli, as Gbx1-/- mice displayed increased latency suggesting reduced pain in the hot plate test (thermosensory functions)”. Should reflect that this is only true in one sex. There are several instances of similar reporting throughout the manuscript.
5. An issue also remains as to the need to discuss how a defect in dorsal spinal neurons could interfere with the proprioceptive afferent system which, as the authors state, projects to the ventral and intermediate spinal cord, not the dorsal horn which was examined in this study. The explanation :“we cannot exclude abnormal functions of those afferents caused by reduced GABAergic cell number in superficial dorsal horn (see below), which is the transition and/or adjacent region for proprioceptive afferents. Such proprioceptive-like deficits may be also secondary to a defect in sensory perception or expression of such sensory response, as indicated by sensory deficits in Gbx1-/- mice and abnormal development of the dorsal horn, the main target of sensory afferents.” . I don’t find the explanation convincing, and perhaps adding support for such mechanism from the literature might help make your case. Regardless, the statement needs to be amended for accuracy:
a. This study has provided evidence for a reduction in the proportion (not number) of GABA neurons (also please correct other references to number of GABAergic cells elsewhere in the manuscript, eg, in page 1 and 12, please check for other instances.)
b. The data analysis failed to show significant sensory deficits in most sets of data. Other than the latency in the beam test in females (proprioceptive test) all other measures for proprioceptive defects were NS (This also applies to the sentence: “Importantly, abnormal gait resemble proprioceptive-like deficits, which was further suggested by deficits in beam-walking, the test used for evaluation of proprioceptive functions”) and “This abnormal gait, documented by a series of behavioral tests, is not due to deficits in muscle strength or motor coordination, but may be related to proprioceptive deficits suggested by reduced performance in beam walking, a test used in studies of proprioception.” Please check for other possible similar isntances.
c. The projection patterns of proprioceptive afferents were also described as normal, so it is unclear why a case is made for a defect in proprioceptive function due to possible measured GABA deficits in a region that in not recipient of those proprioceptive afferents. Perhaps this argument could be strengthened by providing evidence from the literature that might support this case.
d. For the most part development of the spinal cord was normal (based on the description of Nissl staining, and several markers: Lbx1, Lmx1b, Netrin-1, Drg11, etc.) other than the GABAergic phenotype
e. Similarly, “This abnormal gait, documented by a series of behavioral tests, is not due to deficits in muscle strength or motor coordination, but may be related to proprioceptive deficits suggested by reduced performance in beam walking, a test used in studies of proprioception. (in the conclusion)” needs to accurately reflect the results from the proprioceptor tests.

Other:

6. Please amend the materials and methods which says that the histological analysis was done in 3 sections per animal, in 3 animals of each genotype since some of your figures the legend states that the number of animals is 2, not 3. The N for the results of Figure S3 are not provided.
7. Panel D in Figure 8 appears to be flipped horizontally
8. Please provide the molarity and pH of all buffers used.
9. “unmasking in citrate 0.1 M (pH 6)” do you mean citrate buffer?
10. What were the solutions used to dilute the primary antibodies?
11. “Furthermore, electromyography (EMG) measurements revealed that the sensory nerve conduction” Do you mean SNCV?

Other suggestions:
1. Please amend the statement: ” In contrast to our observations, those mutants show disorganized peripherin expression, together with a decrease of Islet1-expressing cells in the ventral horn of the lumbar spinal cord (Buckley et al. 2013).” See Figures 9 (showing change at E14.5-15.5) and Fig 5 (no change at E11.5). Please specify that the statement is correct for ages comparable to yours (not for younger embryos).
2. The sentence “At E12.5, the expression domains of Gbx1 and Gbx2 overlap, both being expressed in the ventricular and mantle zones of the dorsal spinal cord (Rhinn et al. 2004; Waters, Wilson & Lewandoski 2003). As Gbx2 expression is downregulated fter E12.5, both genes are only transiently coexpressed in dorsal spinal cord progenitor cells, and Gbx1 is the only Gbx gene persistently expressed during later dorsal horn development (John, Wildner & Britsch 2005).” Which appears under the hindbrain heading would more appropriately be placed in the next section under the spinal cord heading.
3. I am also confused as to the statement " This may reflect an abnormal development or survival of GABAergic neurons, which in consequence coud lead to abnormal control of neuronal network in dorsal horn, possibly affecting inhibitory circuits throughout the spinal cord." that is presented in the context of the GABA reduction – since the authors go on to show in the following paragraph that cell death as a factor could be ruled out and instead a change of phenotype to glutamate was proposed as a more likely explanation.
4. The sentence "Proprioceptive neuron afferents form two types of termination zones, one in the intermediate spinal cord and one in the ventral spinal cord where they are directly connected to motoneurons (Brown, 1981)" which was added in page 17: I think a similar description of the targets of proprioceptive afferents should be added to the description of afferents in the first paragraph of the Introduction where the rest of the afferent projections to the SC are described.

---

## Round 0.5 · accepted · Accept

I would like to thank the authors for their cooperation through the processing of the manuscript. The article represents a valuable contribution to the field.

There are a few very small editorial changes that I will ask the authors to make prior to final publication:

1. Page 3, last sentence of first paragraph.
The sentence: “Proprioception is mediated by sensory neurons that project through the dorsal spinal cord to an intermediate zone which in turn projects to the ventral spinal cord where direct connection is made to motoneurons (Brown, 1981; for review: Caspary & Anderson 2003)”
Consider changing to: “ […] to the ventral spinal cord where [a] direct connection is made [with or onto] motoneurons […]”
The sentence also needs a period at the end.

2. Page 6, second paragraph:
A period appears to be missing before the sentence starting ”The presence of a wild-type allele was detected using …”

3. Page 6 last paragraph:
The molarity of PBS is expressed as PBS 1x – I still don’t know what that is – please include the actual molarity at this stage, (eg PBS 1x (XXXX M, pH 7.5) and then continue using 1x thereafter if desired.

4. Page 9 first sentence:
“(rpm) in 5 min.” I am not sure but should this be “over” 5 min?

5. The ages at which the gait was analysed is confusing – It is first said that it is analysed at 10 wk of age (p. 8), then at 4-6 wk of age (p.11) and then the onset is described at P16-17 (p.19). To avoid confusion, please specify the age used for the data in Table 1, Figure 2 and supplementary movie (eg add age in the figure legend and into the table).

6. Page 8, First paragraph, second/third line
The statement “and mouse anti-islet …” Suggest changing [and] and substituting with [for] – am assuming the different antibodies were done in different sections and not in the same section, which is what “and” seems to suggest.

7. Figure 5 and Figure 5 legend:
The abbreviation BN is used for “white noise” (assuming you mean “blanche”?) – I suggest changing this to WN, since English speakers might confuse this with Brown Noise (which is what the abbreviation BN is usually used for).